# Cervical ripening in prolonged pregnancies by silicone double balloon catheter *versus* vaginal dinoprostone slow release system: The MAGPOP randomised controlled trial

Caroline Diguisto[1,2,3]*, Amélie Le Gouge[4], Chloé Arthuis[5], Norbert Winer[5], Olivier Parant[6], Christophe Poncelet[7,8], Celine Chauleur[9,10], Jacob Hannigsberg[11], Guillaume Ducarme[12], Denis Gallot[13], Rene Gabriel[14], Raoul Desbriere[15], Gael Beucher[16], Cyrille Faraguet[17], Helene Isly[18], Patrick Rozenberg[19,20], Bruno Giraudeau[2,4], Franck Perrotin[1,2,21], for the Groupe de Recherche en Obstétrique et Gynécologie (GROG)¶

1 Pôle de gynécologie obstétrique, médecine fœtale, médecine et biologie de la reproduction, centre Olympe de Gouges, CHRU de Tours, Tours, France, 2 Université de Tours, France, 3 Université de Paris, CRESS, INSERM, INRA, Paris, France, 4 INSERM CIC1415, CHRU de Tours, Tours, France, 5 Department of Obstetrics and Gynecology, University Hospital of Nantes, NUN, INRAE, UMR 1280, PhAN, Université de Nantes, France, 6 Pôle de gynécologie obstétrique, hôpital Paule-de-Viguier, CHU de Toulouse, Toulouse, France, 7 Department of Obstetrics and Gynecology, Rene DUBOS Hospital, Cergy-Pontoise, France, 8 Université Paris 13, Sorbonne Paris Cité, UFR SMBH, Bobigny, France, 9 Department of Gynecology and Obstetrics, University Hospital of Saint-Etienne, Saint-Etienne, France, 10 INSERM, SAINBIOSE, U1059, Dysfonction Vasculaire et Hémostase, Université Jean-Monnet; CIC1408, Saint-Etienne, France, 11 CHU Brest, Hôpital Morvan, service de gynécologie-obstétrique, Brest, France, 12 Department of Obstetrics and Gynecology, Centre Hospitalier Departemental, La Roche sur Yon, France, 13 Pôle femme et enfant, CHU Estaing, 1, place Lucie-et-Raymond-Aubrac, Clermont-Ferrand, France, Team Translational approach to epithelial injury and repair, UMR6293 CNRS-Université Clermont Auvergne, U1103 Inserm, GReD, Clermont-Ferrand, France, 14 Service de Gynécologie-Obstétrique, Hôpital Maison Blanche, Reims Cedex, Université de Reims Champagne Ardennes, France, 15 Hôpital Saint Joseph, Department of Obstetrics and Gynecology, Marseille, France, 16 Service de gynécologie obstétrique et médecine de la reproduction, CHU de Caen, Caen, France, 17 Service de Gynécologie Obstétrique, Centre Hospitalier de Chartre, France, 18 Service de Gynécologie Obstétrique, Centre Hospitalier Universitaire de Rennes, France, 19 Department of Obstetrics and Gynecology, Poissy-Saint Germain Hospital, Poissy, France, 20 Versailles St-Quentin University, research unit EA 7285. Montigny-le-Bretonneux, France, 21 INSERM U1253 Imaging and Brain (iBrain), Tours, France

¶ Membership of the Groupe de Recherche en Obstétrique et Gynécologie is provided in the Acknowledgements.
* carolinediguisto@gmail.com

**Data Availability Statement:** Data cannot be shared publicly because we wish to control who

## Abstract

### Background

Prolonged pregnancies are a frequent indication for induction of labour. When the cervix is unfavourable, cervical ripening before oxytocin administration is recommended to increase the likelihood of vaginal delivery, but no particular method is currently recommended for cervical ripening of prolonged pregnancies. This trial evaluates whether the use of mechanical cervical ripening with a silicone double balloon catheter for induction of labour in prolonged pregnancies reduces the cesarean section rate for nonreassuring fetal status compared

has access to the data. Data are available from the CHU Tours Institutional Data Access / Ethics Committee (contact via dpo@chu-tours.fr) for researchers who meet the criteria for access to confidential data. The data underlying the results presented in the study are available from dpo@chu-tours.fr.

**Funding:** MAGPOP was funded by the French Ministry of Health (Programme Hospitalier de Recherche Clinique 2015, PHRCN-2015). The funder had no role in study design, data collection, data analysis, data interpretation or writing of the report.

**Competing interests:** The authors have declared that no competing interests exist.

**Abbreviations:** CI, confidence interval; FHR, fetal heart rate; OR, odd ratio; PROM, prelabour rupture of membranes.

with pharmacological cervical ripening by a vaginal pessary for the slow release of dinoprostone (prostaglandin E2).

## Methods and findings

This is a multicentre, superiority, open-label, parallel-group, randomised controlled trial conducted in 15 French maternity units. Women with singleton pregnancies, a vertex presentation, $\geq$41+0 and $\leq$42+0 weeks' gestation, a Bishop score <6, intact membranes, and no history of cesarean delivery for whom induction of labour was decided were randomised to either mechanical cervical ripening with a Cook Cervical Ripening Balloon or pharmacological cervical ripening by a Propess vaginal pessary serving as a prostaglandin E2 slow-release system. The primary outcome was the rate of cesarean for nonreassuring fetal status, with an independent endpoint adjudication committee determining whether the fetal heart rate was nonreassuring. Secondary outcomes included delivery (time from cervical ripening to delivery, number of patients requiring analgesics), maternal and neonatal outcomes. Between January 2017 and December 2018, 1,220 women were randomised in a 1:1 ratio, 610 allocated to a silicone double balloon catheter, and 610 to the Propess vaginal pessary for the slow release of dinoprostone. The mean age of women was 31 years old, and 80% of them were of white ethnicity. The cesarean rates for nonreassuring fetal status were 5.8% (35/607) in the mechanical ripening group and 5.3% (32/609) in the pharmacological ripening group (proportion difference: 0.5%; 95% confidence interval (CI) −2.1% to 3.1%, $p = 0.70$). Time from cervical ripening to delivery was shorter in the pharmacological ripening group (23 hours versus 32 hours, median difference 6.5 95% CI 5.0 to 7.9, $p < 0.001$), and fewer women required analgesics in the mechanical ripening group (27.5% versus 35.4%, difference in proportion −7.9%, 95% CI −13.2% to −2.7%, $p = 0.003$). There were no statistically significant differences between the 2 groups for other delivery, maternal, and neonatal outcomes. A limitation was a low observed rate of cesarean section.

## Conclusions

In this study, we observed no difference in the rates of cesarean deliveries for nonreassuring fetal status between mechanical ripening with a silicone double balloon catheter and pharmacological cervical ripening with a pessary for the slow release of dinoprostone.

## Trial registration

ClinicalTrials.gov NCT02907060.

## Author summary

### Why was this study done?

- Induction of labour may be necessary when pregnancies reach 41 weeks of gestation.

- Ripening methods include mechanical and pharmacological options and both are currently used in pregnancies that reach 41 weeks of gestation.

- Mechanical cervical ripening leads to less uterine tachysystole and less fetal heart rate anomalies than pharmacological methods.

- Whether mechanical methods are associated with reduced perinatal morbidity in prolonged pregnancies in comparison with pharmacological methods needed to be investigated by a sufficiently powered randomised trial.

### What did the researchers do and find?

- We conducted a trial to compare cervical ripening with a silicone double-balloon catheter (mechanical method) to cervical ripening with a vaginal dinoprostone slow-release system (pharmacological method) among women with prolonged pregnancies.

- We did not find that cervical ripening with a silicone double-balloon catheter was superior to cervical ripening with a vaginal dinoprostone slow-release system to reduce the rate of cesarean for nonreassuring fetal heart rate and overall maternal and neonatal morbidity.

### What do these findings mean?

- There is no evidence to justify preferring mechanical cervical ripening over pharmacological cervical ripening in pregnancies that have reached 41 weeks.

## Introduction

Pregnancies that reach 41 weeks are associated with increased rates of oligohydramnios, meconium-stained fluid, fetal heart rate anomalies, cesarean delivery, asphyxia, and perinatal death [1–7]. In many countries, induction of labour is recommended starting at 41 weeks to reduce this mortality and morbidity [8]. When the cervix is unfavourable, cervical ripening before oxytocin administration is recommended to increase the likelihood of vaginal delivery [9]. Ripening methods include pharmacological methods (misoprostol and dinoprostone) and mechanical options (single or double balloon catheters) [10], and the 2 strategies have proved to be equally effective to achieve vaginal deliveries. However, pharmacological ripening appears to lead to more uterine tachysystole causing fetal heart rate (FHR) anomalies and intensive care admission of neonates than mechanical ripening [11–14]. Pharmacological methods being associated with an increased risk of FHR anomalies and suspicion of fetal asphyxia, they may not be the most suitable method when pregnancies reach 41 weeks, as there is already a higher risk of asphyxia in such cases. We hypothesised that mechanical cervical ripening, which involves less uterine hyperstimulation with FHR anomalies, might be associated with better perinatal outcomes in prolonged pregnancies. Comparison of mechanical and pharmacological methods without defining a single device or a single substance among the mechanical and pharmacological techniques available seemed unsatisfactory for reasons of both organisation and interpretation. To standardise interventions, we chose the silicone double balloon catheter for the mechanical group and the Propess system for vaginal slow release of dinoprostone for the pharmacological group.

## Methods

### Study design

MAGPOP is a multicentre, superiority, open-label, randomised controlled trial with 2 parallel groups, which compares mechanical ripening with a silicone double balloon catheter with pharmacological ripening with a pessary for the slow release of dinoprostone among women with prolonged pregnancies. The protocol has been published and is available online (https://bmjopen.bmj.com/content/7/9/e016069.long) [15].

### Participants

Women were recruited in 15 hospital obstetric units (tertiary and nontertiary hospitals) in France all equipped with maternal and neonatal intensive care units. Women were eligible for the study if they were 18 years old or more, had a singleton pregnancy, a fetus in vertex presentation with a term between 41 weeks + 0 days and 42 weeks + 0 (estimated from an ultrasound between 11 and 14 weeks), and had agreed with their obstetric professional that their labour should be induced. The exclusion criteria for the study were a Bishop score ≥6, severe pre-eclampsia, a previous cesarean delivery or other uterine surgery, low-lying placenta, suspected genital herpes infection, known HIV seropositivity, suspected severe congenital abnormalities, pathological FHR, or prelabour rupture of membranes (PROM).

Eligible women were screened from 41 weeks' gestation during the monitoring provided by midwives, sonographers, or physicians, recommended every 2 days at this term by French guidelines [16]. Women were informed of the aims and scopes of the study, and written informed consent was obtained from those who met all inclusion and no exclusion criteria.

### Randomisation and blinding

Women were randomly assigned (1,1 ratio) to mechanical cervical ripening with a silicone double balloon catheter or pharmacological cervical ripening with a pessary for the slow release of dinoprostone. Randomisation and concealment were managed by a secured online centralised web-based system. Randomisation was stratified for centre and parity (nulliparas versus others) and generated by use of permuted blocks of variable size. The sequence was generated by someone who was not involved in patient recruitment. The nature of the intervention made it impossible to blind either the women or obstetric staff. To compensate for the absence of blinding, the primary outcome was adjudicated by an adjudication committee, blinded to the allocation group.

### Interventions

Both interventions were used according to their manufacturer's recommendations. The mechanical cervical ripening device was the Cook Cervical Ripening Balloon (Cook Medical Europe, Limerick, Ireland, reference J-CRBS-184000). A speculum was inserted to obtain cervical access, and the cervix wiped with a solution compatible with the woman's allergies (povidone iodine or sodium hypochlorite). The catheter was introduced so that both balloons reached the extra-amniotic space. Clinicians were recommended to use saline to inflate both balloons. Recommendations were to inflate the upper, uterine balloon (40 ml) first, then gently pull the catheter until the upper balloon abutted the internal os and then inflate the vaginal balloon (40 ml) so that balloons were situated on either side of the cervix. Saline was then inserted in both balloons to a maximum volume of 80 ml per balloon.

For the pharmacological cervical ripening procedure, the clinician inserted the Propess system pessary for the slow release of 10 mg dinoprostone (prostaglandin PGE2 Propess, Ferring SAS, Gentilly, France) against the cervix with or without a speculum.

## Follow-up

Monitoring and management were identical in both groups. After cervical ripening began, FHR was monitored by external tocography for 2 hours [18]. Fetal well-being and uterine activity were then monitored intermittently. Women were admitted to the labour ward if labour started. Epidural analgesia was placed for women who requested it, according to the usual medical indications and contraindications. If labour had not started 24 hours after cervical ripening began, the device was removed to initiate labour induction with oxytocin/amniotomy regardless of cervical status. Oxytocin perfusion did not start until at least 30 minutes after device removal. Induction of labour with oxytocin followed French national guidelines for this procedure [17].

## Setting

Obstetrics professionals (midwives and physicians) responsible for inserting the devices were accustomed to using both options in their daily practice as participating obstetrics hospital units all used both the silicone double balloon catheter and the Propess vaginal delivery system regularly. A meeting was held in each participating unit before inclusions began to ensure homogeneity of practices for the use of the 2 devices. The 2015 FIGO classification for fetal heart rate interpretation was also reviewed with onsite physicians during this meeting to ensure homogeneous interpretation of FHR between different maternity units [18].

## Outcomes

The **primary outcome** was the **cesarean section rate for nonreassuring fetal status,** with this status determined by a blinded adjudication committee. This independent committee comprised 3 physicians with a high level of expertise in FHR interpretation. When inclusions were over and all the data collected, members of the committee adjudicated the primary outcome, by independently reviewing the 2 hours of fetal heart rate preceding the cesarean section for all women with cesarean deliveries to determine if nonreassuring fetal status was the main indication for the cesarean. The 3 committee members discussed the cases for which they had discordant opinions to reach a consensus about whether or not the fetal heart rate was reassuring.

Among the **secondary outcomes** (which were not adjudicated) were those related to delivery, including time from cervical ripening to delivery in hours and delivery rates within 12 and 24 hours after cervical ripening began, need for induction with oxytocin, total oxytocin dose before delivery, uterine hyperstimulation defined as more than 6 contractions per 10 minutes over any 30-minute period, hyperstimulation treated by tocolysis, uterine rupture, suspicious or pathological FHR, and use of analgesics and antibiotics. The obstetricians performing the cesarean deliveries reported they were indicated for nonreassuring fetal status (in their own opinion, before the adjudication committee's decision) or for other reasons, which were specified. The type of vaginal delivery (spontaneous or instrumental) and the indication for operative vaginal deliveries were reported. The cesarean delivery rate was added as a secondary outcome after the registration of the trial on clinicaltrials.gov.

**Outcomes related to maternal morbidity** were suspected maternal intra- or postpartum infection, postpartum haemorrhage defined as estimated blood loss >500 ml and blood transfusion. Perineal complications and admission to intensive care were added as secondary outcomes after the registration of the trial on clinicaltrials.gov.

**Outcomes related to neonatal morbidity** included the Apgar score (presented only at 5 minutes, the timing most relevant for prognosis), umbilical arterial pH at delivery, admission to a neonatal unit or an intensive care unit, respiratory distress with need for any respiratory support, and neonatal asphyxia. Suspected neonatal infection was added as a secondary outcome after the registration of the trial on clinicaltrials.gov.

## Sample size calculation

We hypothesised that the cesarean rate for nonreassuring FHR would be 17.7% in the pharmacological group (pessary for the slow release of dinoprostone) [19] and that mechanical cervical ripening (silicone double balloon catheter) would reduce the rate to 12%, which was considered to be a sufficient difference to justify recommending this method. With a power of 80% and a 2-tailed type I error of 5%, this would require to include 1,220 women (610 in each group).

## Statistical analysis

A statistical analysis plan was finalised before the database was frozen. Statistical analyses were performed according to the intention-to-treat principle. Baseline characteristics were reported per group with numbers and percentages for categorical variables and with means and standard deviation and medians and interquartile ranges for continuous variables. For the primary outcome, missing data were managed by simple imputation (as multiple imputation was not possible): Women were considered to have had a cesarean for nonreassuring FHR. Rates were then compared with the chi-squared test. The between-group difference in proportions was estimated as well as its 95% confidence interval (CI) (Wald method). Results were also presented as crude and adjusted odd ratios (ORs) with their 95% CIs; the adjusted OR was estimated from a random logistic regression model to take into account the centre effect. Secondary outcomes were analysed by the $\chi^2$ or Fisher exact test, as appropriate, for qualitative data and by the Student or Wilcoxon test for quantitative data. The difference in the proportion or median was estimated for each secondary outcome, along with its associated 95% CI. Statistical analyses were performed with SAS 9.4 and R 3.3.1 software.

## Ethics and dissemination

The study protocol was approved by the competent authorities (Agence Nationale de Sécurité du Médicament et des produits de santé and Comité de Protection des Personnes de TOURS —Region Centre; 2016-R23, 29/11/2016) and registered at ClinicalTrials.gov (NCT02907060) and in the European EudraCT database (2016-A00952-49).

## Results

Between January 27, 2017, and December 23, 2018, 8,850 women were assessed for eligibility in 15 maternity units (S1 Data), 1,700 met the inclusion and exclusion criteria, and 1,220 were randomised (S1 Fig). Due to 2 revocations of consent ($n = 2$) and 2 issues with consent forms ($n = 2$), 1,216 women were analysed: 607 in the mechanical cervical ripening group (silicone double balloon catheter) and 609 in the pharmacological group (pessary for the slow release of dinoprostone). In the end, the primary outcome was missing for 2 cases in the mechanical group (which prevented us from doing multiple imputations).

Baseline characteristics are described in the S1 Table. Cervical ripening with a silicone double balloon catheter was not associated with a lower rate of cesareans for nonreassuring fetal status (as determined by the adjudication committee): 35/607 (5.8%), compared with 32/609

(5.3%) with the pessary for the slow release of dinoprostone (crude OR 1.10, 95% CI 0.67 to 1.81; the difference in proportions was 0.5%, 95% CI −2.1% to 3.1%, $p = 0.70$). No maternity unit effect was observed (adjusted OR for maternity unit effect 1.10, 95% CI 0.75 to 1.61). A complete-case analysis observed no significant difference between the 2 groups (5.5% in the silicone double balloon catheter group compared versus 5.3% in the pessary for the slow release of dinoprostone group, difference in proportions 0.2%, 95% CI −2.3% to 2.7%, $p = 0.88$). Other outcomes related to cervical ripening and delivery are reported in S2 Table. Time from cervical ripening to delivery was shorter in the pharmacological ripening group (23 hours versus 32 hours, median difference 6.5 95% CI 5.0 to 7.9, $p < 0.001$), and fewer women required analgesics in the mechanical ripening group (27.5% versus 35.4%, difference in proportion −7.9%, 95% CI −13.2% to −2.7%, $p = 0.003$). Maternal and neonatal morbidity did not differ significantly between the groups (S3 and S4 Tables). Three neonatal deaths occurred: 1 case of an unknown bilateral diaphragmatic hernia not identified antenatally, 1 extensive necrotizing volvulus, and 1 case of severe neonatal hypoxia complicated by intestinal perforation.

## Discussion

Our results did not show that mechanical cervical ripening with a silicone double balloon catheter was associated with a lower rate of cesarean delivery for nonreassuring fetal status in pregnancies that reached 41 weeks. It was associated with a lower rate of analgesic use and of delivery within 24 hours and with a greater need for oxytocin in the second phase of induction.

MAGPOP is, to our knowledge, the first multicentre randomised trial to compare mechanical and pharmacological cervical ripening in pregnancies that reach 41 weeks, and its results are consistent with those from previous meta-analysis [13,14].

Our trial has several strengths. First, our inclusion of a large population from heterogeneous settings in different French regions in both university and general hospitals with public and private units that differ in size ensures strong external validity. A second strength lies in the choice of the rate of cesarean delivery for nonreassuring fetal status for the primary outcome as we wanted an outcome measure for both mother and fetus, or at least implying both maternal and fetal morbidity, without using a composite outcome. Although these outcomes are often used in obstetrics to enable significant results to be obtained with fewer patients, they are usually composed of an aggregate of endpoints of varying importance and are accordingly difficult to interpret [20,21]. A third strength is that the primary outcome has been adjudicated. Although cesarean delivery is an objective measure, the decision to perform a cesarean is not objective. Physicians often disagree about the need for cesarean delivery [22], and physicians facing the same situation twice may make different decisions each time: Our primary outcome is thus one that is "objectively measured but potentially influenced by clinician judgment" [23]. To avoid bias due to this physician influence on outcome, especially in view of the impossibility of blinding them to the intervention, a blinded independent committee adjudicated the primary outcome. The final strength of the trial is the stratification of the randomisation by centre, which helped to limit bias due to possible differences between units in practices for oxytocin management and cesarean indications [24].

The main limitation of the trial is that the observed rate of cesarean section for nonreassuring fetal status in the group treated pharmaceutically was much lower than expected. As a consequence, the a priori specified 5.7 percentage point difference (i.e., from 17.7% to 12%) appears debatable a posteriori. Nevertheless, the trial is large with 1,216 women included in the analysis, and point estimates are very close (5.8% versus 5.3%) and higher in the mechanical cervical ripening than in the pharmacological group. Our a priori hypothesis was based on

data from a trial conducted between 2009 and 2012 among women with prolonged pregnancies undergoing pharmaceutical cervical ripening in French maternity units, some of which took part in this trial [19]. In that earlier trial, the authors observed a global cesarean rate fairly similar to ours. The rate indicated for nonreassuring fetal status here may have been lower in part because this trial was conducted 7 years later, and practices for the use of oxytocin have changed over the years in France, with nationwide surveys in 2010 and 2016 showing that physicians tend to use less oxytocin [25]. More appropriate management of uterine hyperstimulation with more frequent use of tocolysis during labour over the years may also explain the changes in indications for cesareans. Otherwise, we note that the rates of cesarean deliveries for nonreassuring fetal status determined by the adjudication committee were lower than those reported by the obstetricians at the delivery. This may be explained by the level of expertise for FHR interpretation of members of the committee, combined with the anxiety that may be associated with being on call and facing the patient but also by the fact that the committee's information for making these judgements was essentially limited to the 2 hours of fetal rate recordings before the decision to perform a cesarean.

## Conclusion

Mechanical ripening with a silicone double balloon catheter was not found to be superior to pharmacological cervical ripening with a pessary for the slow release of dinoprostone in reducing the rate of cesarean deliveries for nonreassuring fetal status in prolonged pregnancies. This result does not provide evidence that justify preferring 1 strategy over the other.

## Supporting information

**S1 Data. List of participating maternity units.**
(DOCX)

**S1 Fig. Flow chart of eligibility, randomisation.**
(TIFF)

**S1 Table. Baseline characteristics of the intention to treat population.**
(DOCX)

**S2 Table. Outcomes related to cervical ripening and delivery.**
(DOCX)

**S3 Table. Outcomes related to maternal morbidity.**
(DOCX)

**S4 Table. Outcomes related to neonatal morbidity.**
(DOCX)

**S1 CONSORT Checklist.**
(DOCX)

## Acknowledgments

The authors thank women who participated in the study and members of the adjudication committee Charles Garabedjan, Muriel Doret, and Philipe Deruelle.

Participating members/collaborators of the Groupe de Recherche en Obstétrique et Gynécologie (GROG) were Thomas Schmitz, Elie Azria, Catherine Deneux-tharaux Anne Ego, François Goffinet, Cyril Huissoud, Gilles Kayem, Bruno Langer, Camille Le Ray, Olivier

Morel, Marie-Victoire Senat, Loïc Sentilhes, Damien Subtil, and Christophe Vayssiere. Participating members of the GROG had a role in the design of the study. They have not received any compensation for their role in the study.

They also wish to thank the research team from the University Hospital of Tours, which included Celine Lengelle, Renaud Respaud, Aurélie Darmarillacq, Yoann Desvignes Martine Gardin, and Elie Guichard.

## Author Contributions

**Conceptualization:** Caroline Diguisto, Amélie Le Gouge, Chloé Arthuis, Bruno Giraudeau, Franck Perrotin.

**Data curation:** Bruno Giraudeau, Franck Perrotin.

**Formal analysis:** Amélie Le Gouge.

**Funding acquisition:** Caroline Diguisto, Bruno Giraudeau.

**Investigation:** Caroline Diguisto, Chloé Arthuis, Norbert Winer, Olivier Parant, Christophe Poncelet, Celine Chauleur, Jacob Hannigsberg, Guillaume Ducarme, Denis Gallot, Rene Gabriel, Raoul Desbriere, Gael Beucher, Cyrille Faraguet, Helene Isly, Patrick Rozenberg.

**Methodology:** Caroline Diguisto, Franck Perrotin.

**Project administration:** Caroline Diguisto, Franck Perrotin.

**Supervision:** Caroline Diguisto, Bruno Giraudeau, Franck Perrotin.

**Validation:** Caroline Diguisto, Amélie Le Gouge, Bruno Giraudeau.

**Visualization:** Caroline Diguisto.

**Writing – original draft:** Caroline Diguisto, Bruno Giraudeau, Franck Perrotin.

**Writing – review & editing:** Caroline Diguisto, Amélie Le Gouge, Chloé Arthuis, Norbert Winer, Olivier Parant, Christophe Poncelet, Celine Chauleur, Jacob Hannigsberg, Guillaume Ducarme, Denis Gallot, Rene Gabriel, Raoul Desbriere, Gael Beucher, Cyrille Faraguet, Helene Isly, Patrick Rozenberg, Bruno Giraudeau, Franck Perrotin.

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
