## [Editor Report · Decision Letter 0]

9 Jun 2020

Dear Dr Diguisto, 

Thank you for submitting your manuscript entitled "Cervical ripening in prolonged pregnancies by silicone double balloon catheter versus vaginal dinoprostone slow release system: the MAGPOP randomised controlled trial" for consideration by PLOS Medicine.

Your manuscript has now been evaluated by the PLOS Medicine editorial staff as well as by an academic editor with relevant expertise and I am writing to let you know that we would like to send your submission out for external peer review.

Kind regards,

Artur Arikainen,

Associate Editor

PLOS Medicine

---

## [Decision Letter · Decision Letter 1]

15 Jul 2020

Dear Dr. Diguisto,

Thank you very much for submitting your manuscript "Cervical ripening in prolonged pregnancies by silicone double balloon catheter versus vaginal dinoprostone slow release system: the MAGPOP randomised controlled trial" (PMEDICINE-D-20-02567R1) for consideration at PLOS Medicine. 

[LINK]

In light of these reviews, I am afraid that we will not be able to accept the manuscript for publication in the journal in its current form, but we would like to consider a revised version that addresses the reviewers' and editors' comments. Obviously we cannot make any decision about publication until we have seen the revised manuscript and your response, and we plan to seek re-review by one or more of the reviewers. 

We expect to receive your revised manuscript by Aug 05 2020 11:59PM. Please email us (plosmedicine@plos.org) if you have any questions or concerns.

We look forward to receiving your revised manuscript. 

Sincerely,

Emma Veitch, PhD

PLOS Medicine

On behalf of Clare Stone, PhD, Acting Chief Editor,

PLOS Medicine

plosmedicine.org

*Please structure your abstract using the PLOS Medicine headings (Background, Methods and Findings, Conclusions - "Methods and Findings" should be a single subsection). 

*Ideally, in the last sentence of the Abstract Methods and Findings section, please include a brief note about any key limitation(s) of the study's methodology.

*At this stage, we ask that you include a short, non-technical Author Summary of your research to make findings accessible to a wide audience that includes both scientists and non-scientists. The Author Summary should immediately follow the Abstract in your revised manuscript. This text is subject to editorial change and should be distinct from the scientific abstract. Please see our author guidelines for more information: https://journals.plos.org/plosmedicine/s/revising-your-manuscript#loc-author-summary

*The paper includes a CONSORT flowchart showing flow and inclusion of participants through the study but it would be good to also note that the reporting of the paper follows CONSORT guidelines (http://www.consort-statement.org/), and include a completed CONSORT checklist as supporting information alongside the revised paper when resubmitting. (When using the CONSORT checklist, of course update any relevant sections of the paper with more detailed methodological information as recommended by the reporting guideline). 

*One reviewer queried whether the protocol for the trial was publicly available, but we noted that the published BMJ Open paper corresponding to the protocol was cited in the submitted paper, so perhaps they just missed this (ref 16, cited Methods section). 

Comments from the reviewers:

Reviewer #1: This is a well conducted trial. It is large. It was registered prospectively. The primary outcome "Caesarean for fetal compromise" was prespecifed and is reported. Compliance and follow up is good. The lack of blinding is a slight weakness but unavoidable with these two methods. 

Comments 

The result is that there was no differecne between the methods. This is in accord with the latest systematic review. The present trial is the largest reported but adds little to that review. 

I note the conclusions of the Cochrane review. "Low- to moderate-quality evidence shows mechanical induction with a balloon is probably as effective as induction of labour with vaginal PGE2. However, a balloon seems to have a more favourable safety profile. More research on this comparison does not seem warranted. [...] Future research could be focused more on safety aspects for the neonate and maternal satisfaction." 

This trial does not so far as i can see report maternal satisfaction. However it does compare the balloon with the slow release dinoprosotne system (propess) which is aguably the safest form of prostaglandin. It also adds to the safety profile of the balloon catheter which is valuable. 

The primary outcome is somewhat subjective, although an adjudication committee was used. As a safety check it would have been wise to report total caesarean rates to ensure that any differences were not just due to reclassifaction. 

I note that Total caesarean rate was not a secondary outcome. However it is reported. This is good. 

Could the authors check that the secondary outocmes reported align with those prespecified on trial registry. If addtional outcomes are reported this should be noted. 

Minor points

1. remove the row multiparous in table 1 . it is redundant because the classes are mutually exclusive 

2. for prepregnancy weight, height and BMI remove the median IQR rows. Mean (SD) is better 

3. remove mean (SD) apgar score in table 4. Ditto mean and median UA pH.

4. Please remove duplicated results for text and tables. Remove them from the text. 

5. Please highlight the primary outocme in the relevant results table. 

Jim Thornton. Nottingham 19 June 2020

Reviewer #2: This is a manuscript that provides an important contribution to the literature on method of IOL. 

IOL is an area of interest in maternity care. With increasing rates of IOL, changing indications for IOL and different settings for initiating IOL (home vs hospital) this manuscript is likely to be of interest.

The trial compares two methods of cervical ripening: balloon catheter and Prostaglandin slow release pessary. Both methods are commonly used in high income settings (although WHO also supports misoprostol and foley catheter as alternatives - both of which are cheaper)

The primary outcome is caesarean birth for indicated by non-reassuring fetal heat rate tracing (suspected fetal distress) as judged by three independent clinicians review of fetal monitoring with treatment allocation masked (blinded evaluation of primary outcome).

It is a large (1220 women randomised) multicentre RCT according to a published protocol which are strengths. 

Comments:

1. The main limitation is that the primary event rate was only 1/3 of that anticipated, so it is underpowered to detect differences in this. 

- The conclusion that "Mechanical ripening was not superior to pharmacological cervical ripening in reducing the rate of caesarean deliveries for non-reassuring fetal status in prolonged pregnancies" cannot be made on this evidence, and should be tempered and discussed. 

-Is there any way of including discussion of, given the extremely similar primary outcomes, how big a trial would have to be to detect differences/the likelihood of there being clinically meaningful differences?

2. A problem is interpretation the results. Based on MAGPOP, and the underpowered primary outcome. I expect clinicians will use MAGPOP to support their preconceived views. Prostaglandin enthusiasts will quote the findings that labour is quicker, with less oxytocin and less failure to progress. Balloon catheter enthusiast will quote lower uterine hyperstimulation, less analgsia and clinician reported fetal distress. Discussion of results in the context of other RCTs may help -for example PROBAAT (Jozwiak M, Oude Rengerink K, Benthem M, et al. Foley catheter versus vaginal prostaglandin E2 gel for induction of labour at term (PROBAAT trial): an open-label, randomised controlled trial. Lancet. 2011;378(9809):2095-2103. doi:10.1016/S0140-6736(11)61484-0). Probaat had a very similar CS rate to MAGPOP trial, with both showing no difference in CS rate overall. Probaat also saw a statistically significant reduction in uterine hyperstimulation with mechanical methods. Could meta-analysis of MAGPOP findings with other trial findings be considered as part of the paper? Or some kind of presentation of pros and cons in cases per 1000 (for every 1000 IOL there would be ..) to help women and clinicians make decisions about method of IOL. 

3. Were any secondary outcomes adjudicated? For example, in view of clinician prior belief that PG are associated with hyperstimulation, there may have been a lower threshold to report and treat uterine hyperstimulation in the PG arm than the Ballon arm. If not adjudicated this should be discussed. Also, it sounds like only CS were adjudicated. Is it possible that different proportions of vaginal births 'should' have been CS as indicated by objective review of fetal heart rate monitoring?

4. The amount of missing data should be clearly stated, including the level of primary outcome data "imputed according to the worst case scenario method (assuming that women with missing data in the mechanical group had a caesarean for. non-reassuring FHR and that those in the pharmacological group did not). " As imputation of data was not stated in the protocol, this decision should be highlighted as being post hoc. If levels of missing data were high a sensitivity analysis of compete case data could be considered. 

5. Generalisability to different settings should be discussed, in light of the protocols used. 

6. I wasn't clear if clustering by centre was considered and how this was handled in analysis.

7. Minor points: some clarifications would be useful e.g. "a gynaecologic position for its

Insertion" - not sure what this is. " cervix wiped with a solution compatible with the woman's allergies to prepare for device insertion" - what are the options for solutions?

Overall, I believe this manuscript should be published with i) more measured discussion of the limitations of the trial and ii) interpretation of findings in the context of other evidence. 

Data availability should be encouraged for transparency of the current analyses, and to allow meta-analysis with other studies to estimate effects with more precision - especially for some important secondary outcomes.

Reviewer #3: This is a well conducted trial, on an important subject. The researchers have compared two strategies for induction of labour in a group of women with a low Bishop score, to test the hypothesis that the use of a mechanical device (in comparison with prostaglandin) reduces Caesarean section due to abnormal FHR patterns during induction of labour. The results will be of interest and clinical use for clinicians and patients.

The study is well designed and executed. The choice of best primary outcome in 'induction' trials is the subject of debate. The choice of the authors here seems sensible, and is well justified. As mentioned, it is not possible to mask either caregivers or participants, but the use of independent outcome assessment is to be commended. Retention in the trial is good. The participants are representative of a relevant population. The exclusion criteria of Caesarean section and women of < 41 weeks gestation are important - some comment in the discussion about whether the results of the trial would be generalisable to these groups might be helpful. 

Although there was no difference the rate of the primary outcome between the groups, there are some important differences in secondary outcomes, which will be relevant to caregivers and pregnant women. I agree is not possible to say which treatment is "better" - some pregnant women will prioritise the lower rate of analgesia in the balloon catheter group, others will prioritise the speed of induction and the lower time to delivery in the prostaglandin group. Although one should be careful about over interpreting differences between secondary outcomes, it is appropriate to highlight the differences here - they are unlillely to be a Type 1 error.

I think this might be a revision (although I haven't seen any previous versions). There is very little to improve on in this manuscript. Some minor comments that the authors may wish to consider as they prepare the final version of this manuscript are as follows:

1. Please confirm both interventions were used according to the manufacturer recommendations (or describe the deviations if they were not).

2. The study was a superiority trial but the trial did not show superiority. It would be worth a line or two about the impact of this on the conclusion that "there is no difference between the two treatments"

3. Please describe in more detail the justification for planning a primary outcome rate of 12% in the cervical balloon group

4. It might also be worth mentioning that hyperstimulation higher in prostaglandin group.

5. Please comment on generalisability as mentioned above.

6. Is the detailed protocol published on line somewhere - if so a link would be helpful.

Reviewer #4: Thanks for the opportunity to review your manuscript. My role is as a statistical reviewer so my comments and queries are focused on the design, analysis, and data from the study. 

This is a very clear and concise manuscript, and the study design has gone to all means possible to blind study staff where possible (e.g. endpoint adjudications). 

One aspect I felt I didn't clearly understand was the generalisation from mechanical vs. pharmacological treatments. There is a hypothesis that from differences in uterine activity that mechanical treatments could be superior in avoiding caesarean deliveries. This seems reasonable to me (as a non-expert), but I couldn't follow why these two particular treatments were used, and whether they could be broadly seen as representative of other mechanical and pharmacological treatments that are available. It would be safer to specifically refer to these particular types of treatments rather than to the broad categories of 'pharmacological' vs. 'mechanical', and I am interested to see what the reviewers with expertise in this area think of this. 

There were 17 secondary outcomes listed in the clinicaltrials.gov registration. How was it decided which of these would be included in this main paper? 

Was a detailed statistical analysis plan developed for this study before database lock/unblinding? 

There are a very small number of (4 in total) reported as not analysed, what is the number (per treatment arm) who were analysed but didn't have the main or secondary outcomes? Could this information about missingness be included a supplementary table?

The main analysis used a single-imputation strategy (caesarean for mech group, no caesarean for pharm group). This is reasonable approach, and if two more scenarios (oppose to worst-case scenario, and non-responder imputation (assume all missing patients had caesareans) are included and show similar results this would demonstrate that the results are robust to any assumptions about missing data. 

What strategy was used for secondary outcomes with missing data?

There are specific comments below but an overall comment overall was that I couldn't see which specific tests applied to which of the secondary outcome variables (e.g. in Table 2 footnotes could be included). I would also prefer to see an effect estimate (HR, OR, Mean difference) with a 95% CI presented for the study outcomes in the tables and throughout the abstract and main body.

P4, Sites. 

What was the proportion of patients recruited at each of the sites?

P4, Randomisation.

What were the parity strata (no previous/previous)? 

Does 'varying sizes' mean random block size? What were the random block sizes used?

P7. Outcomes related to maternal morbidity.

Were these outcomes added before DB lock and unblinding?

P8. Analysis

Should this be 'random intercept logistic regression'? 

What was the criteria for using Chi-squared or Fisher? Observed or expected cell size?

Fisher's exact test will be conservatively biased in a 2x2 table which doesn't have fixed margins, Barnard's test is a more powerful alternative when the margins are not fixed. There are also exact regression models that area also good alternatives where 

What type of analysis was done for time to delivery? Were all deliveries completed within the 2 days of the time-frame detailed in the clinicaltrials.gov? Typically this would be done with a survival analysis where censoring could be accounted for.

P9. Results. 

Could the time to delivery be presented in Kaplan-Meir plots (in supplementary appendix if space is an issue)?

The sample size calculation assumed that caesarean section rate would be 17.7% and 12% in the two study arms, while the overall rate was closer to 5%. Was overall event-rate monitored during the study? 

P10. While useful, randomisation stratified by centre is fairly typical for an RCT and I don't necessarily view this as a strength of study design.

[LINK]

---

## [Decision Letter · Decision Letter 2]

9 Oct 2020

Dear Dr. Diguisto,

Thank you very much for re-submitting your manuscript "Cervical ripening in prolonged pregnancies by silicone double balloon catheter versus vaginal dinoprostone slow release system: the MAGPOP randomised controlled trial" (PMEDICINE-D-20-02567R2) for review by PLOS Medicine.

I have discussed the paper with my colleagues and the academic editor and it was also seen again by 3 reviewers. I am pleased to say that provided the remaining editorial and production issues are dealt with we are planning to accept the paper for publication in the journal.

[LINK]

We look forward to receiving the revised manuscript by Oct 16 2020 11:59PM. 

Sincerely,

Artur Arikainen

Associate Editor 

PLOS Medicine

plosmedicine.org

Requests from Editors:

1. Please address any final reviewer comments.

2. The Data Availability Statement (DAS) requires revision. If the data are not freely available, please include an appropriate contact (web or email address) for inquiries - this cannot be a study author.

3. Please include line numbers in the margin throughout.

4. Abstract:

a. Please delete this sentence: “The a priori hypothesis was that mechanical ripening would reduce this caesarean rate from 17.7 to 12%.”

b. Please rename “main outcome” as “primary outcome” here and throughout.

c. Please include summary participant demographics (eg. age, ethnicity).

d. Please include descriptions and results for secondary outcomes.

e. Please present a summary of safety data for the study including numbers of events and whether or not adverse events are thought to be related to treatment.

f. In the last sentence of the Abstract Methods and Findings section, please include a brief note about any key limitations of the study's methodology, including limited C-section rates.

g. Conclusions: Begin with “In this study, we observed that…”

h. Delete this sentence: “The difference in point estimates is 0.5 percentage points, in favour of the Propess group.”

i. Rather than "the trial failed to demonstrate ...", please state "there was no difference [in the primary outcome]".

5. Author Summary: Please expand your author summary to include the following 3 subheadings, and one or two bullet points of text for each: Why Was This Study Done?, What Did the Researchers Do and Find?, What Do These Findings Mean?. Please see our author guidelines for more information (with an example): https://journals.plos.org/plosmedicine/s/revising-your-manuscript#loc-author-summary

6. Please use the "Vancouver" style for reference formatting, and see our website for other reference guidelines https://journals.plos.org/plosmedicine/s/submission-guidelines#loc-references. Citations should not be superscript. The reference list should not use bold/italics, and the first 6 authors should be shown before “et al.”

7. Methods:

a. Please list the participating maternity units here or in Supplementary Information.

b. Please give the recruitment dates, including day/month/year.

c. Please clarify whether consent was informed.

8. Results: Please include secondary outcomes.

9. Discussion: Please change as follows: “MAGPOP is, to our knowledge, the first multicentre randomised trial to compare…”

10. Page 14-15: Please remove Data availability, Authors' contributions, Funding statement and Competing interests statement – these are taken from the online submission form.

11. Tables: Please ensure all abbreviations are defined in footnotes, eg. “SD”, “BMI”, “CI”.

12. Please report p values as p<0.001, where lower than this limit.

13. When completing the CONSORT checklist, please use section and paragraph numbers, rather than page numbers.

---

Comments from Reviewers:

Reviewer #1: This is a second revision. I've lready reviewed this and the authors have responded. I'm not sure why I'm looking at it again. Time for the editors to do soem work. Accept. 

Reviewer #2: Happy with revisions. No further comments to add.

Reviewer #4: Thanks for the revision and detailed replies to my comments. I am happy with the responses to my original comments, the changes made resolved my queries and I think have strengthened the work. I can happily published with one minor suggestion.

One of the other reviewers asked to see the primary outcome (cesarean rate) in a table - I think this a good idea, as it should make it easier for readers to find the primary result.

I agree with the authors in their comment about the powering - the CI for the main outcome provides the most useful guidance for interpreting the results in light of a lower than expected event rate and is interpreted appropriately by the authors.

[LINK]

---

## [Editor Report · Decision Letter 3]

13 Jan 2021

Dear Dr. Diguisto,

I am writing concerning your manuscript submitted to PLOS Medicine, entitled “Cervical ripening in prolonged pregnancies by silicone double balloon catheter versus vaginal dinoprostone slow release system: the MAGPOP randomised controlled trial.”

We have now completed our final technical checks and have approved your submission for publication. You will shortly receive a letter of formal acceptance from the editor.

Kind regards,

PLOS Medicine